Advancements in monitoring: a comparison of traditional and application-based tools for measuring outdoor recreation

Vilalta Capdevila Talia 1
McLellan Brynn A. brynn@y2y.net 1
Loosen Annie 1 2
Forshner Anne 3
Pigeon Karine 1 2 4 5
Jacob Aerin L. 1 2 6
Wright Pamela 2
Ehlers Libby 1
1 Yellowstone to Yukon Conservation Initiative , Canmore , Alberta , Canada
2 University of Northern British Columbia , Prince George , British Columbia , Canada
3 Banff, Yoho and Kootenay National Parks, Parks Canada , Lake Louise , Alberta , Canada
4 British Columbia Ministry of Water, Lands, and Resource Stewardship, Government of British Columbia , Smithers , British Columbia , Canada
5 IUCN SSC Bear Specialist Group
6 Nature Conservancy of Canada , Victoria , British Columbia , Canada
Ghermandi Andrea
Electronic publication date: 2024 Sep 10
Publication date: 2024
Volume: 12
Electronic Location ID: e17744
Received 2024 Mar 14; Accepted 2024 Jun 24
Copyright: ©2024 Vilalta Capdevila et al.
Copyright year: 2024
Copyright holder: Vilalta Capdevila et al.
License: This is an open access article distributed under the terms of the Creative Commons Attribution License, which permits unrestricted use, distribution, reproduction and adaptation in any medium and for any purpose provided that it is properly attributed. For attribution, the original author(s), title, publication source (PeerJ) and either DOI or URL of the article must be cited.
License URL: https://creativecommons.org/licenses/by/4.0/

Keywords: Aerial surveys, Camera traps, Participatory mapping, Wildlife and recreation management, Recreation monitoring, App-based data, User-generated data, Volunteered geographic information, Trail counters

Funding: The Animal Welfare Institute, Donner Canadian Foundation No. E-792 109-23 Habitat Conservation Trust Foundation, RBC Foundation No. ENV20228402 793 Volgenau Foundation, and the Wilburforce Foundation No. YELLO2311 This work was supported by the Animal Welfare Institute, Donner Canadian Foundation (No. E-792 109-23), Habitat Conservation Trust Foundation, RBC Foundation (No. ENV20228402), 793 Volgenau Foundation, and the Wilburforce Foundation (No. YELLO2311). There was no additional external funding received for this study. The funders had no role in study design, data collection and analysis, decision to publish, or preparation of the manuscript.

==============================
Outdoor recreation has experienced a boom in recent years and continues to grow. While outdoor recreation provides wide-ranging benefits to human well-being, there are growing concerns about the sustainability of recreation with the increased pressures placed on ecological systems and visitor experiences. These concerns emphasize the need for managers to access accurate and timely recreation data at scales that match the growing extent of the recreation footprint. Here, we compare spatial and temporal patterns of winter and summer recreation using traditional (trail cameras, infrared counters, aerial surveys, participatory mapping) and application-based tools (Strava Metro, Strava Global Heatmap, Wikiloc) across the Columbia and Canadian Rocky Mountains of western Canada. We demonstrate how recreation use can be estimated using traditional and application-based tools, although their accuracy and utility varies across space, season and activity type. We found that trail cameras and infrared counters captured similar broad-scale patterns in count estimates of pedestrians and all recreation activities. Aerial surveys captured areas with low recreation intensity and participatory mapping captured coarser information on the intensity and extent of recreation across large spatial and temporal scales. Application-based data provided detailed spatiotemporal information on recreation use, but datasets were biased towards specific activities. Strava Metro data was more suited for capturing broad-scale spatial patterns in biking than pedestrian recreation. Application-based data should be supplemented with data from traditional tools to identify biases in data and fill in data gaps. We provide a comparison of each tool for measuring recreation use, highlight each tools’ strengths and limitations and applications to address real-world monitoring and management scenarios. Our research contributes towards a better understanding of which tool, or combinations of tools, to use that can expand the rigor and scope of recreation research. These findings support decision-making to mitigate pressures on wildlife and their habitats while allowing for high-quality recreation experiences.

Introduction

Interest in outdoor recreation and visitation to parks, protected and non-protected areas is increasing globally, with terrestrial protected areas receiving approximately eight billion visits annually (Balmford et al., 2015; Outdoor Industry Association, 2017). Outdoor recreation (hereafter, recreation) provides benefits to human well-being (Hartig et al., 2014; Lackey et al., 2021). However, there are growing concerns about the sustainability of recreation with the pressures that increasing recreation places on ecological systems (Larson et al., 2016; Schulze et al., 2018; Granados et al., 2023) and impacts to recreation experiences. Wildlife responses to recreation vary widely and are context and species specific (Larson et al., 2016; Granados et al., 2023). Yet, recreation has been identified as the most common threat to species at risk in Canada (Rosenthal et al., 2022) and has been linked to reduced species richness, diversity and abundance (Sato, Wood & Lindenmayer, 2013; Larson et al., 2019), shifts in wildlife habitat use (Naidoo & Burton, 2020; Granados et al., 2023; Gump & Thornton, 2023), and loss and degradation of wildlife habitat (Heinemeyer et al., 2019). Simultaneously, the rise in recreation has led to crowding and conflicts between different user groups, reducing the quality of recreation experiences for many (Manning & Valliere, 2001; Jackson, Haider & Elliot, 2003; Wolf, Brown & Wohlfart, 2018; Ferguson et al., 2023).

As the demand for recreation grows, managers are increasingly faced with making decisions to protect plant and wildlife communities while supporting high-quality recreation experiences (Loosen et al., 2023). These decisions often necessitate understanding spatial and temporal patterns in recreation (Loomis, 2000) and the impact of recreation on ecological systems, including wildlife use patterns. Yet, an understanding of where, when, how intense, and what activity types occur remains difficult to quantify. The lack of accessible and current recreation data is consistently highlighted as a critical knowledge gap impeding managers’ decision-making and strategic planning (Loomis, 2000; Columbia Mountains Institute of Applied Ecology, 2023; Loosen et al., 2023). This hinders progress towards sustainable recreation (Hadwen, Hill & Pickering, 2007), especially for unpredictable and off-trail recreation in remote areas (Braunisch, Patthey & Arlettaz, 2011).

Here we focus on two categories of tools for measuring and monitoring recreation: traditional and application-based tools. We define traditional tools as any tool used specifically to count and monitor recreationists, and application-based tools as any tool that provides data from an activity sharing platform where geographic information is voluntarily shared by the app user (Goodchild, 2007). To achieve research, monitoring and management goals it requires an understanding of the advantages and limitations of each tool and their associated data (Cessford & Muhar, 2003; Norman & Pickering, 2017; McCahon, Brinkman & Klimstra, 2023). Understanding the differences between traditional and application-based tools has been recognized as a research priority (Norman & Pickering, 2017). While several studies have incorporated multiple tools to measure recreation (Norman & Pickering, 2017; Fisher et al., 2018; Wood et al., 2020), comparison of tools across large, continuous landscapes (i.e., inside and outside protected areas) and multiple recreation activities and seasons remains relatively unexplored.

Traditional monitoring tools can include access permits, voluntary self-reporting or registration (Fredman et al., 2009), aerial surveys (Heinemeyer et al., 2019), on-site counts and surveys of people on trails or cars in parking lots (Fisher et al., 2018), participatory mapping (PM; Brown & Weber, 2011; Brown & Kyttä, 2014), trail cameras, and infrared counters (Pettebone, Newman & Lawson, 2010; Miller, Leung & Kays, 2017; Naidoo & Burton, 2020). Many of these tools provide detailed and contextual information on the type of recreation activity and the number, behavior or characteristics of recreationists. Compliance of self-reporting permits and registration can vary due to a lack of enforcement and lead to underestimates of recreation use (Fredman et al., 2009). On-site survey methods often require prior information on where people are recreating, making it challenging to capture new areas (e.g., where people use trails or areas not managed for recreation; Vilalta Capdevila et al., 2022; Lawson, 2021). Alternatively, managers and researchers can use PM to identify areas with recreation. PM engages participants to identify recreation areas or the intensity of recreation activities on a map (Brown & Weber, 2011; Rösner et al., 2014; Wolf et al., 2015; Wolf, Brown & Wohlfart, 2018; Komossa, Wartmann & Verburg, 2021). While PM can provide insight into the presence and spatial distribution of recreation and recreationists’ behavior, cognitive bias can limit the accuracy and inference to specific times and locations (Brown, 2011). Other tools such as trail cameras (hereafter cameras) and infrared counters (hereafter, counters) passively collect spatiotemporal data (Lawson, 2021). Installing and maintaining large camera and counter networks across extensive landscapes can be logistically challenging and require substantial resources. Yet, large-scale studies are often needed to capture expanding recreation footprints and to characterize spatial and temporal recreation patterns at an extent appropriate for decision-makers (e.g., beyond a single protected park or trail network). In these instances, traditional tools are often inadequate for collecting large-scale recreation data.

In contrast to traditional tools, application-based tools capitalize on large, crowdsourced datasets. These tools take advantage of the fitness and recreation data generated by the proliferation of mobile devices and global positioning systems (GPS), helping to overcome the challenges of many traditional tools (Wilkins, Wood & Smith, 2021). Generally, the primary purpose of application-based tools are centered around the user’s performance or fitness rather than recreational research or monitoring. Application-based tools like Strava, AllTrails, and Wikiloc collect geographical information when recreationists upload their activities (Norman & Pickering, 2017; Lee & Sener, 2020; Goodbody et al., 2021; Ferguson et al., 2023; Venter et al., 2023). Application-based tools link an individual’s activity track, photograph, or point location to a specific time and geographic coordinate. While application-based tools can generate high volumes of data across large extents and over extended periods (Fisher et al., 2018), analyses of these complex datasets often require technical expertise, extensive computational processing and storage (Lee & Sener, 2020; Wood et al., 2020) and consideration of privacy, ethical use, and financial costs. Further, application-based tools can also be spatially and temporally biased in the type of users, geography, and data available (Norman & Pickering, 2017; Venter et al., 2023). Despite the promise of application-based tools for measuring recreation, these types of data are only beginning to be explored.

There is a need to understand the different tools and data sources to provide clear recommendations for evidence-based decision-making. We compared data from traditional and application-based tools to measure spatial and temporal patterns of summer (May 1–Oct 31) and winter (Nov 1–Apr 30) recreation use. We used data from four traditional tools (counters, cameras, aerial surveys, PM) and three application-based tools (Strava Metro, Strava Global Heatmap, Wikiloc) in western Canada. This area is a hub for motorized and non-motorized recreation with growing concerns about managing conflicts between recreation use (Holterman, Wright & Jacob, 2023) and maintaining high-quality wildlife habitat. We explore the relationships between recreation data from tools over time, space, and activity type (Table 1). First, we compare recreation data across the study area to examine broad use patterns across tools. Second, we use case studies to assess smaller areas and trail network-specific use patterns. Collectively, this work provides a unique opportunity to compare data from multiple tools that span a range of recreation intensities, activities, and human development and ecological conditions.

Table 1 Research questions that compare data from traditional and application-based tools.

Research question	Tools compared	Recreation type	Season 1	
Study area	
1A & 1B	How do monthly counts of recreation use correlate across space (1A) and time (1B?).	Strava Metro2, cameras, counters	All activities, biking, pedestrian	Year-round 	
2	How do annual counts of recreation use correlate across space?	Strava Heatmap3, Strava Metro2, cameras, counters	All activities  	Year-round 	
Case study	
1	How does the spatial distribution and intensity of summer motorized recreation use compare across space?	Wikiloc, PM	Motorized	Summer	
2	How does the spatial distribution and intensity of winter motorized recreation use compare across space?	Aerial surveys, Wikiloc, PM	Motorized	Winter	
3	How do monthly counts of biking recreation use correlate across space?	Strava Metro2, cameras, counters	Biking	Year-round	
Notes.

1 Season dates: summer (May 1–Oct 31) and winter (Nov 1–Apr 30).

2 Strava Metro contains counts of recreation users by activity type (biking, pedestrian) and is only available to Strava Metro partners.

3 Strava Heatmap contains aggregated annual indices of recreation (run, hike, walk, ride, water, winter) and is publicly available.

PM: abbreviation for participatory mapping.

Methods

Portions of this text were previously published as part of a prepint Vilalta Capdevila et al. (2024).

Study area

The study area encompasses 63,000 km2 of mountainous terrain in western Alberta and eastern British Columbia, Canada, and contains 20,260 km2 (32%) of federal and provincial protected areas, including provincial and federal parks, wilderness areas, and heritage rangelands (Fig. 1; Loosen et al., 2023). The study area includes open public and private lands, grazing leases, motorized (e.g., public land use zones in Alberta) and non-motorized recreation areas (e.g., National Parks). Commercial snowmobiling and heli-assisted activities occur in some areas that are not serviced by maintained roads. The study area includes the population centers of Canmore, Banff, Golden, Revelstoke and Nakusp (Fig. 1).

Figure 1 Study area (A) and case study areas (B, C, D).

Case studies include the (B) Ghost Public Land Use Zone (PLUZ) that has a high density of motorized trails in a non-protected area, (C) the Kootenay mountains that has winter motorized recreation, and (D) biking trails surrounding the town of Canmore in the Bow Valley. Base map sources: Esri, TomTom, Garmin, FAO, NOAA, USGS, ©OpenStreetMap contributors, and the GIS User Community.

The study area is within the traditional territories of the Okanagan/Syilx, Sinixt, Ktunaxa, Secwépemc, Niitsitapi (Blackfoot) Nations of Siksika, Kainai, Piikani, and Aamskapi Pikuni; the Îyârhe (Stoney) Nakoda Nations of Goodstoney, Bearspaw, and Chiniki; Tsuut’ina First Nation; Mountain Cree. It also includes lands within Treaties 6, 7, and 8 and districts 1, 3 and 4 of the Otipemiskiwak Métis Government of Alberta. The study is part of the greater Yellowstone to Yukon region that includes key linkages for ecological connectivity, core protected areas and human modified landscapes (Holterman, Wright & Jacob, 2023). This region includes alpine, sub-alpine and montane subregions and habitat for wildlife sensitive to human presence such as grizzly bears (Ladle et al., 2018), wolverines (Heinemeyer et al., 2019; Barrueto et al., 2022) and caribou (Seip, Johnson & Watts, 2007; Wilson & Wilmshurst, 2019).

Resource developments are prominent across the study area and includes forestry, mining and oil and gas. Forestry and mining operations contribute to polygonal footprints and resource roads (Forest Practices Board, 2021). The study area includes 53,436 km of trails, gravel resources roads, and cutlines, pipelines, and transmission lines with an overall density of 0.85 km/km2 (Loosen et al., 2023). Roads and trails associated with resource and industrial developments provide access for motorized and non-motorized recreation (Ministry of Forests, Lands, and Natural Resource Operations, 2013; Government of Albert 2024; Forest Practices Board, 2021).

Measuring recreation

Traditional tools

Infrared counters – We acquired data from TRAFx infrared trail or vehicle counters (TRAFx Research Ltd., Canmore, Alberta, Canada) from partners (provincial and federal government, researchers, and conservation organizations). Counters were positioned along trails and roads to count the number of people or vehicles that pass through the device’s infrared scope. Details on counter models and set up are provided in supplemental information (Data Description S1).

In ArcMap (ArcMap Version 10.8.2, ESRI, Redlands, California) we retained data from trails only (Vilalta Capdevila et al., 2022) and filtered by activity type. A portion of these devices count bicycles (vehicle counters with bike mode activated) and we considered any data from bike counters to represent bike counts only, while data from the other counters were assumed to represent pedestrian counts. TRAFx trail counters do not differentiate between recreation activity types, and we could not validate if any other activities were counted. To represent recreation intensity, we aggregated daily count data from 2017 to 2019 to monthly totals per year for pedestrian, biking, and all recreation activities.

Trail cameras – We obtained remote trail camera datasets from provincial and federal government and researcher partners. Cameras were deployed on trails and low-traffic roads to monitor wildlife but incidental wildlife data was also collected. Camera images were classified by the number of recreationists and recreation activity type. To represent recreation intensity, we aggregated daily counts of recreationists from 2017 to 2019 to monthly totals per year and split the counts in three categories: pedestrian (any foot-based activity) and biking (road, mountain, electric and fat bikes), and all activities (all motorized and non-motorized activities). Details on activity types, camera models, set up, and image classification are provided in Data Description S1.

Aerial surveys – We conducted aerial surveys to quantify the intensity and spatial extent of on and off trail winter recreation in three survey areas (Data Description S1; Fig. 2). We completed two rounds of surveys in a helicopter between February and April 2022, with each survey occurring at least a month apart. Following Heinemeyer et al. (2019), we organized the survey into 1.5 × 1.5 km (2.25 km2) grids (Data Description S1). We flew transects, that were spaced 3 km apart, with two observers seated on either side of the helicopter. We recorded the type of recreation (snowmobiling and backcountry, heli and cat skiing) and the spatial footprint of recreation tracks within a 1.5 × 1.5 km observation window. We classified the footprint into five bins based on the percent cover of recreation tracks: <10, 10–25, 26–50, 51–75, ≥76%). Raw data were processed to convert observation waypoints into a polygon grid representing the observation window; see Data Description S1 for further details.

Figure 2 Spatial distribution of data from (A) counters, cameras, Strava Metro and (B) public participatory mapping, aerial surveys, and Wikiloc within the study area (gray boundary).

Strava Heatmap not displayed (see Data Description S1). Base map sources: ©OpenStreetMap and contributors, CC-BY-SA.

Participatory mapping – We conducted participatory mapping (PM) interviews with recreation experts (e.g., park rangers, recreation group, trail users, lodge and campground owners) across our study area (Fig. 2). Meetings with 36 participants were conducted from 2020 to 2021 over Zoom (Zoom Video Communications, Inc. 2020, San Jose, California United States). We shared a digital map on the screen and asked participants to identify areas with recreation and the types of activities and estimated average number of people per day recreating in these areas from 2017 to 2019. Participants marked locations on the shared map. We categorized the estimated daily number of people recreating in an area into three bins: 1–10; 11–50; >50 recreationists per day. Recreation areas were digitized to a 1.5 × 1.5 km cell raster. Each cell was assigned one of the three bin values to represent recreation intensity. In areas where participants indicated differing numbers of recreationists in the same place, we used the highest estimate. See Data Description S1 for further details.

Application-based tools

Strava Metro – Strava Metro (https://metro.strava.com) uses non-motorized biking and pedestrian (running, walking, hiking) GPS data collected from users’ phones, fitness watches or GPS devices. Strava Metro snaps GPS tracks to the closest OpenStreetMap (http://www.openstreetmap.org) street and trail segment. OpenStreetMap is a crowdsourced geographical database of the world. Strava Metro data are anonymized to maintain user privacy by aggregating count data for trail segments with at least three unique users per direction (up or down the activity segment), representing totals as multiples of five (Strava Metro 2019, pers. comm.). Strava Metro provides data as hourly, daily, monthly, or annual counts. We aggregated the number of users in each direction along a trail segment from 2017 to 2019 into monthly totals per year for biking, pedestrians and all activities. We acquired a Strava Metro partnership through Alberta and British Columbia government ministries.

Strava Global Heatmap – Strava Global Heatmap (hereafter, Strava Heatmap; https://www.strava.com/heatmap) is a crowdsourced heatmap that uses GPS-based recreation tracks of public non-motorized recreation activities (run, hike, walk, ride, water, winter) recorded on phones, fitness watches, and GPS devices. Strava Heatmap is freely available, unlike Strava Metro, and has been used as an index of human recreation (Jäger, Schirpke & Tappeiner, 2020; Corradini et al., 2021; Carlson et al., 2022). The heatmap displays annual aggregated indices of public activities for each calendar year, with updates to the heatmap occurring monthly. The Strava Heatmap is an accumulation of individual user tracks and uses a cumulative distribution function to normalize the intensity of “heat” that maximizes contrast. To avoid visual artifacts in areas of low use or large gradients of use, bi-linear interpolation is applied (Robb, 2017). Roads and trails with little use are not displayed on the heatmap until several tracks are uploaded. We used the 2016 to 2017 heatmap and converted the raster to a greyscale gradient such that the color of each cell represented an index of recreation intensity. We used the ‘greyscale’ image analysis function in ArcGIS Pro (Version 3.1.0) to produce a grid of high and low use intensity areas (max = 255, min = 0; Data Description S1).

Wikiloc – Wikiloc (https://www.wikiloc.com) uses GPS-based recreation tracks recorded on phones, fitness watches, and GPS devices that users can upload to a public website. Wikiloc data includes a range of motorized and non-motorized recreation activities. We downloaded motorized tracks (quad, snowmobile, off-road vehicle, motorbike) available within our study area (Fig. 2). We saved the GPS tracks, recreation type, time and date for each track and removed unique personal identifiers.

Analyses

Comparison of monthly counts across time and space

We correlated counts from cameras, counters, and Strava Metro across space and time (questions 1A and 1B; Table 1). Datasets were acquired from multiple research projects and therefore were not always aligned spatially (Fig. 2) or temporally (Table 2). To match tools, devices were paired when cameras and counters were within 200 m of each other along the same trail or were within 30 m of a Strava Metro trail segment and had data for the same day. To match cameras and counters, we used trails to create a distance raster between devices and selected distances within 200 m and data from the same day. We selected the distance threshold of 200 m so that each tool was likely to capture the same recreationist along a trail, while maximizing the number of matched locations. We used 30 m to match the Strava Metro trail segments to the camera or counter because Strava Metro data are already linked to OpenStreetMap trail segments and the buffered distance used to create the trail network dataset used in this work was 30 m (Vilalta Capdevila et al., 2022). We refer to matched pairs of tools as spatially matched locations. We grouped analyses into all recreation activity types, pedestrian, and biking. We did not include comparisons of motorized activities due to data sparsity.

Table 2 Comparison of tools to monitor recreation use.

The table includes a description of how each tool quantifies recreation use, data type (resolution), minimum temporal grain, temporal and spatial extent, recreation activity type (motorized, non-motorized, all recreation activity types), season (summer, winter, year round).

Tool	How recreation is quantified	Data type	Temporal grain	Temporal extent	Spatial extent 2	Activity	Season 3	
Counters	Number of recreationists per unit time	Point	Daily	2017–2019	Portion	All activities	Year round	
Cameras	Number of recreationists per unit time	Point	Daily	2017–2019	Portion	All activities	Year round	
Aerial surveys	Percent cover of recreation tracks	Grid (1,500 × 1,500 m)	Daily	Feb–Apr, 2022	Portion	All activities	winter	
PM 1	Estimated average number of recreationists in an area	Grid (1,500 × 1,500 m)	Annual	2017–2019	Portion	All activities	Year round	
Strava Metro	Number of trail segments per unit of time	Linear	Daily	2017–2019	Full study area	Non-motorized	Year round	
Strava Heatmap	Index of recreation intensity	Grid (19 × 19 m)	Annual	2016–2017	Full study area	Non-motorized	Year round	
Wikiloc	Number of trail segments per unit of time	Linear	Daily	2013–2023	Portion	Motorized	Year round	
Notes.

1 Participatory mapping (PM).

2 See Fig. 2 for a map of the spatial distribution of the different tools.

3 Season dates: summer (May 1–Oct 31) and winter (Nov 1–Apr 30).

To compare counts between tools across space, we calculated Pearson’s correlation values for the monthly counts (i.e., sum of all counts within a month) for each spatially matched location using data pooled across three years (2017–2019). To compare counts between tools over time, we calculated Pearson’s correlation for the monthly counts pooled across all spatially matched locations and for each of the three years (2017, 2018, 2019). For both spatial and temporal analyses, we calculated correlations for counts of all activities, pedestrian, and biking. We considered correlation values greater than 0.75 as strong correlation. All analysis occurred in R (Version 4.3.1, R Core Team, 2023).

Comparison of annual counts across space

We compared data from cameras, counters, and Strava Metro to aggregated indices of recreation intensity from Strava Heatmap (question 2; Table 1). We visually compared annual Strava Heatmap index values to median annual counts from Strava Metro and cameras and counters that were deployed for a minimum of nine months (273 days) within a year. We chose nine months as our temporal unit for the cameras and counters to best match Strava Heatmaps temporal scale (i.e., the heatmap represents the annual aggregated indices of public activities) while also including a range of camera and counter devices across the study area because most devices, particularly cameras, did not collect data continuously for a year. For Strava Metro, we compared the heatmap intensity values to the mid-point of the nearest Strava Metro segment. For cameras and counters, we generated a 50-m buffer around each device to match the maximum width of trails and used the highest raster value from Strava Heatmap within the buffer as relative index. This approach accounted for the heatmap which displays high use trails with a wider footprint relative to low use trails.

Results

We recorded 2,030,653 total recreation counts from cameras, 79,775,734 counts from counters, and 28,211,105 counts from 32,374 Strava Metro trail segments. See Table S1 for number of spatially matched locations for all activities, biking and pedestrian activities. Some counters displayed extreme monthly counts for pedestrian and biking activities and all tools displayed generally higher counts in the summer (Fig. S6). We collected 1,673 unique Wikiloc motorized tracks from 2013–2023. PM data was collected from 36 participants (Data Description S1).

Figure 3 Distribution of Pearson’s correlation values of monthly counts of (A) all recreation activities, (B) pedestrian, and (C) biking for pairwise combinations of spatially matched cameras, counters and Strava Metro, 2017–2019.

Each row represents different data comparisons and the dashed red line represents the mean Pearson’s correlation value.

Comparison of monthly counts across space and time

Spatial correlation (research question 1A) – We found high correlation across spatially matched counters, cameras and Strava Metro (minimum r = 0.72 across all comparisons; Fig. 3). Mean correlation was highest for all activities, compared to pedestrian or biking (Fig. 3). Mean correlation values was highest for the cameras and counters for all activities (r = 0.92; Fig. 3) and pedestrians (r = 0.91; Fig. 3). Correlation values of all activities, pedestrians and bikers varied across the study area, with clusters of spatially paired counters, cameras and Strava Metro locations with low correlation (r < 0.5) surrounding the developments of Canmore, Banff and Lake Louise (Fig. S7).

Temporal comparison (research question 1B) – Across all tools, correlation values varied by month and year (Fig. 4). Low correlation values occurred generally during May to August, with this pattern most pronounced for comparisons between Strava Metro and cameras or counters for all activities (Fig. 4). There was generally high year-round correlations for camera and counters of all activities and pedestrians (Fig. 4). Biking displayed more variable correlations across months and years compared to all activities and pedestrians (Fig. 4).

Figure 4 Pearson’s correlation values of monthly counts for (A, D, G) all recreation activities, (B, E, H) pedestrian and (C, F, I) biking for pairwise combinations of spatially matched cameras, counters and Strava Metro, 2017–2019.

Each row represents different data comparisons and the red line represents a non-linear trendline (generalized additive model).

Comparison of annual counts across space

Spatial correlation (research question 2) – High relative Strava Heatmap values coincided with both high and low annual median counts of cameras, counters or Strava Metro (Fig. 5). There were many locations where relative high Strava Heatmap values coincided with low median counts of the three tools, particularly for counters and Strava Metro (Fig. 5). There were also locations where Strava Heatmap indicated no recreation yet cameras and Strava Metro captured low levels of recreation (Fig. 5).

Figure 5 Strava Heatmap index of recreation intensity compared to annual median recreation counts from (A) cameras, (B) counters, and (C) Strava Metro.

Data for cameras, counters and Strava Metro are from 2017 to 2019 for all spatially matched locations in the study area, data from Strava Heatmap is from 2016 to 2017. Trendline represented in red.

Case studies

Comparing summer motorized recreation patterns in a popular recreation area

Assessing the distribution and intensities of off-trail recreation remains a long-standing challenge in land-use and recreation management and planning. While traditional tools such as counters or cameras provide spatiotemporal information of recreation on trails, it can be difficult for these devices to capture recreation across large extents and trail networks with many entry points or where use is dispersed (e.g., backcountry skiing in alpine areas). PM (Brown & Weber, 2011; Wolf et al., 2015; Komossa, Wartmann & Verburg, 2021) and application-based data (Norman & Pickering, 2017; Ghermandi & Sinclair, 2019) offer alternative approaches for collecting recreation data, but it remains unclear how or if the data complements each other.

Figure 6 Summer motorized recreation in the Ghost Public Land Use Zone (PLUZ), Alberta, from May to September, 2017–2019.

(A) Represents estimates of the average number of people recreating between 2017 to 2019 from participatory mapping (PM); gray shading represents areas where PM participants indicated that motorized recreation was present, but no estimates were provided. (B) Displays the density of cumulative motorized Wikiloc tracks (km/km2). (C) Displays Wikiloc tracks (blue) and motorized features (orange) from Vilalta Capdevila et al. (2022). Base map sources: Esri, TomTom, Garmin, FAO, NOAA, USGS, ©OpenStreetMap contributors, and the GIS User Community.

Figure 7 Recreation trails in the Ghost Public Land Use Zone, Alberta, Canada, September 2023.

Photo credit: Talia Vilalta Capdevila, Brynn McLellan.

Here we compare the use of PM and Wikiloc for estimating summer motorized recreation intensity in the Ghost Public Land Use Zone (PLUZ), Alberta, Canada (Figs. 1; 6; 7). This area includes more than 1,500 km2 of public lands east of Don Getty Wildland Provincial Park and Banff National Park (Government of Alberta, 2024), a network of designated and non-designated motorized and non-motorized trails (Yarmoloy & Stelfox, 2011; Weerstra, 2018; Fig. 1). The area is used for forestry, agriculture, oil and gas, and hosts a range of year-round recreation activities. There are several campgrounds and opportunities for un-serviced camping in designated camping nodes and random camping. To evaluate annual counts of summer motorized use between tools, we aggregated binned PM estimates of the number of summer motorized recreationists into 1.5 × 1.5 km grid cells. To match the temporal scale of the PM datasets, we partitioned the Wikiloc tracks from 2017 to 2019 to include motorized activities from May to September. We calculated the total length of tracks within each cell to produce a density (km of tracks/km2) grid in ArcMap.

PM and Wikiloc captured similar spatial extents of recreation, although patterns in recreation intensity varied due to the resolution and user-group biases of the datasets (Fig. 6). Although Wikiloc provided spatiotemporal information of on and off-trail recreation use in remote areas, the data represents the relative recreation use among app users and are a subset of all recreation users (Norman & Pickering, 2017). Here we found that the Wikiloc data did not capture all motorized trails in the Ghost PLUZ (Fig. 6). Comparatively, PM provided information on the spatial extent and the approximate number of recreationists in some areas of the Ghost PLUZ. While estimates of use were not provided for all areas in the Ghost PLUZ, some PM participants indicated that un-designated trails outnumbered designated trails (Fig. 6). The combination of general and area-specific information generated from PM and Wikiloc datasets illustrate how combining these tools could allow for a more robust assessment of recreation. This tool combination could be used to identify areas to more intensively monitor with traditional tools. This approach would be particularly useful in large areas with spatially dispersed or high-density trails systems with many entry points, such as the Ghost PLUZ, where employing and maintaining traditional tools can be time and resource intensive (Wolf, Brown & Wohlfart, 2018).

Comparing winter motorized recreation patterns in the Kootenay Mountains

Backcountry winter recreation is a fast-growing sector, with many mountain towns relying on winter recreation for their local economy (Outdoor Industry Association, 2017). The growing popularity of winter recreation combined with advancements in recreation equipment has allowed recreationists to travel further into remote and undisturbed areas, including sensitive wildlife habitats (Heinemeyer et al., 2019). Recreational activities can displace species from high quality habitats, such as wolverine (Heinemeyer et al., 2019), caribou (Seip, Johnson & Watts, 2007) and mountain goats (Richard & Côté, 2016). Managing winter recreation to allow for the protection of wildlife habitat remains an ongoing challenge, one that is further intensified by the lack of data on dispersed, off-trail winter recreation. It can be difficult to capture dispersed winter-use with traditional tools, such as cameras or counters, because snowpack enables widespread travel either on-or off-trails.

To highlight the challenges of monitoring off-trail winter recreation we compared PM, aerial surveys and Wikiloc for capturing winter motorized recreation near the population centers of Golden and Revelstoke in the Kootenay Mountains, British Columbia (Figs. 1; 8). These population centers are hubs for winter recreation and tourism and rely on recreation for their local economies (Morten & Der, 2019; Tourism Revelstoke, 2019). We filtered the PM and aerial survey datasets to include snowmobiling and the Wikiloc dataset to include motorized activities from November to April, 2015–2023, and visually compared the spatial extent and use intensity of winter motorized recreation activities.

Figure 8 Winter motorized recreation intensity for (A) aerial surveys, (B) Wikiloc linear track density (km/km2), and (C) participatory mapping (PM) in the Kootenay Mountains, British Columbia, Canada.

Aerial surveys represent the percent footprint of snowmobile tracks in each grid cell. Hashed gray shading represents the areas surveyed by aerial flights between February and April 2022. Wikiloc linear track density represents the cumulative density (km/km2) of winter (November–April) GPS tracks from the app-users, 2015–2023. Participatory mapping represents the estimated number of motorized winter recreationists, 2017–2019. Base map sources: Esri, TomTom, Garmin, FAO, NOAA, USGS, ©OpenStreetMap contributors, and the GIS User Community.

All tools provided similar estimates of where winter motorized recreation occurred, although patterns in the spatial extent and intensity of recreation use varied due to the resolution and activity type (Fig. 8). PM identified areas where recreation occurred over the broadest extent, but did not provide continuous or as detailed spatiotemporal information on the number of recreationists as the other tools. Although PM data are low resolution data, the data captured similar patterns as Wikiloc for areas with high winter motorized use. Comparatively, aerial surveys captured areas with snowmobiling that the two other tools did not, particularly in areas with low intensity of use (Fig. 8). This incomplete representation of winter motorized recreation by Wikiloc and PM, particularly in areas with low intensity of use, may be attributed to the application-based data being dependent on the number and types of users of the app and PM participants not being as familiar with less popular winter motorized recreation areas.

While we demonstrate how traditional and application-based tools can collect off-trail winter recreation data, the accuracy of estimates, type of recreation activity, and spatial and temporal resolution varies across tools. Combining multiple tools allows for areas with a range of use intensity to be represented and for more robust estimates of winter motorized recreation. Moreover, this case study underscores the need for further development of tools to accurately monitor dispersed winter recreation. This information is critical to better understanding and managing the impacts of winter recreation on sensitive and snow-associated species (Heinemeyer et al., 2019).

Comparing biking counts from Strava Metro to counters and cameras in a recreation community

The Bow Valley of Alberta is a hub for year-round outdoor and adventure tourism (Alberta Government, 2018). Both humans and wildlife concentrate their activities along valley bottoms. As a result these areas are disproportionately affected by human development that can lead to human-wildlife conflicts, displacement of wildlife and loss of wildlife corridors (Alberta Government, 2018). Mirroring many mountain towns in North America, the town of Canmore has undergone growth in their population, development footprint and recreation sector (Alberta Government, 2018; Carlson et al., 2022), including informal recreation trails (Whittington et al., 2022). While monitoring and research on recreation patterns and impacts to wildlife has been identified as a research priority in the Bow Valley (Carlson et al., 2022), this work is hampered by lack of current and detailed information on recreation. While traditional tools, such as cameras and counters, have been used to monitor recreation in this area, application-based data from fitness tracking apps has been largely unexplored.

Here we illustrate how the application-based tool Strava Metro provides similar estimates of bikers as cameras and counters on trails networks surrounding the town of Canmore. We calculated Pearson’s correlation values for monthly counts of bikers between Strava Metro and cameras or counters for all spatially matched locations from three years of data (2017–2019). Correlations were mapped to identify patterns (Fig. 9).

Figure 9 Pearson’s correlation values of monthly counts of biking activities for Strava Metro compared to (A) cameras and (B) counters in Canmore, Alberta, Canada.

Base map sources: Esri, TomTom, Garmin, Maxar, Airbus DS, NGA, USGS, NASA, CGIAR, N Robinson, NCEAS, NLS, NOAA, OS, NMA, Geodatastyrelsen, Rijkswaterstaat, GSA, Geoland, FEMA, Intermap, ©OpenStreetMap contributors, and the GIS User Community.

Strava Metro captured similar monthly counts of bikers to cameras and counters (Fig. 9); 89% and 77% of the spatially matched locations of Strava Metro and camera or counter locations had high correlations (r ≥ 0.81), respectively (Fig. 9). These findings complements existing research demonstrating that Strava Metro data is generally a good approximation for the spatial extent of biking and use intensity (Lee & Sener, 2020). However, correlation of Strava Metro to biker counts can vary depending on how data are aggregated and are influenced by user-group biases (Lee & Sener, 2020; Venter et al., 2023). Similar to many application-based recreation data, Strava Metro can overrepresent certain user groups (e.g., tech-savvy, middle-aged, fitness-focused ; Venter et al., 2023). Canmore is a hub for recreation tourism and home to many recreation enthusiasts. The high correlation of Strava Metro data to actual biker counts from cameras and counters may be attributed to a large proportion of bikers in the area using the app. However, as suggested by previous research (Venter et al., 2023), Strava Metro may not capture spatial and temporal patterns in biking across all local scales and consideration for the increase in the platforms usership and types of users is also important for long-term analyses (i.e., over several years). Therefore, pairing Strava Metro with other tools, such as counters and cameras, allows for cross-validation and more accurate recreation estimates. For instance, the intensity of biking recreation could be estimated on trails without cameras or counters using Strava Metro and multipliers from other cameras and counter locations. This approach allows for detailed recreation data to be scaled up to match growing recreation footprints and to meet the increasing demand for recreation.

Discussion

The growth of recreation has come with a demand for tools that provide accurate, timely and detailed recreation data (Cessford & Muhar, 2003; Loosen et al., 2023). To make evidence-based decisions, managers and researchers require an understanding of the differences between recreation monitoring tools and data sources. Our study supports research which demonstrates that a combination of traditional and application-based tools expands the rigor and scope of estimating recreation use across large landscapes. However, the accuracy and utility of each tool is context-dependent and varies across space, time and activity type. Cameras and counters captured similar broad-scale patterns in count estimates of all recreation activities and pedestrians. Although application-based data provided detailed spatiotemporal information on recreation, they were biased towards specific recreation activities. For instance, Strava Metro was more suited for capturing broad-scale spatial patterns in biking than pedestrian recreation. Traditional tools including aerial surveys and PM captured the recreation extent and intensity of use. Aerial surveys captured areas with low intensity of use and PM captured recreation information at large spatial and temporal scales. We offer guidelines on monitoring tools, and how to use these tools to address monitoring and management scenarios (see Table S2 for a comparison across the tools).

Correlations between cameras, counters and Strava Metro varied across space, time, and activity type (Figs. 3, 4). Compared to cameras and counters that capture activities regardless of whether a recreationists was using an app, data from application-based tools reflect particular activities and a proportion of the total recreation population (Norman & Pickering, 2017). Previous research indicates that Strava Metro over-represents bikers relative to pedestrians (Venter et al., 2023), and supports our findings where Strava Metro was better correlated with cameras or counters for biking than pedestrian activities across space (Fig. 3). In contrast, we found lower correlation for biking counts between cameras and counters (Figs. 3, 4). This pattern could be attributed to the low sample size (Fig. 3) or counters underestimating or overestimating use. Counters cannot distinguish people in a group and may underestimate groups of recreationists, or overestimate use when counters are triggered by non-target movements such as moving vegetation and animals (Miller, Leung & Kays, 2017; Lawson, 2021; McCahon, Brinkman & Klimstra, 2023). Pairing cameras, that provide spatiotemporal count information on group size and activity types, with counters that provide estimates but not absolute counts, in the same location would allow for counters to be cross-validated and their accuracy assessed.

We found lower correlation clusters for comparisons of Strava Metro to cameras and counters during summer (May–August; Fig. 4) when activity counts peaked (Fig. S6) and for correlations of pedestrian counts surrounding some population centers (Fig. S7). These finding contrasts previous research demonstrating high correlations of Strava Metro and counters across space (Venter et al., 2023). A possible explanation for the trend may be that the Strava Metro data does not represent the total recreation population or volume of recreationists across the entire study area, particularly near some population centers and during the summer. The population centers in our study area are surrounded by high-density trail networks and are hubs for recreation tourism. Strava Metro is typically geared towards more competition and fitness-focused users and has spatial and user-group biases (Venter et al., 2023; also see Lee & Sener, 2020 for review of Strava Metro). Further, Strava Metro does not represent children under the age of 13 (the minimum age to use the app; Strava Terms of Service 2023). Taken together, the lower correlations near some population centers in the summer may be attributed to differences in the behavior between more fitness-focused Strava users who may avoid congested or popular recreation areas, and summer tourists and families that recreate closer to population centers and may be less represented in the apps data.

We found correlation between tools can vary depending on the aggregation of data across time and space. Past studies have found higher correlations between Strava Metro and counter stations across space than time (Venter et al., 2023). This trend may be attributed to Strava Metro not including all data at finer spatial units (i.e., daily) because the privacy threshold requires at least three activities per unit time to be stored on a database (Venter et al., 2023). As a result, monthly or annual aggregation may allow greater number of trail segments to be included, particularly in less popular areas for Strava users.

Strava Heatmap is increasingly used as a proxy for human recreation (Jäger, Schirpke & Tappeiner, 2020; Corradini et al., 2021; Carlson et al., 2022). For example, research demonstrates that Strava Heatmap reflects the intensity of ski mountaineer activities from counting stations (Jäger, Schirpke & Tappeiner, 2020). In contrast, we found that Strava Heatmap did not consistently capture similar patterns as other tools: high recreation counts from cameras, counters and Strava Metro corresponded with both high and low Strava Heatmap index values. Further, cameras and Strava Metro captured recreation in locations where Strava Heatmap indicated no recreation. As Strava Heatmap reflects a proportion of recreation users, it may not capture recreation in less popular or remote areas. We stress the importance of accounting for this limitation, especially when trying to understand interactions between recreationists in remote areas where there are sensitive, wary wildlife species, like wolverines and caribou, who occupy rugged habitats and are sensitive to low levels of human-use (Barrueto et al., 2022). The mismatch across tools may also be attributed to temporal differences among our datasets. The Strava Heatmap dataset represented cumulative recreation data from 2016 to 2017, whereas cameras, counter and Strava Metro data represented recreation use across three years (2017, 2018, 2019). Furthermore, Strava Heatmap is not a straightforward index: map index values are not comparable at large distances because the same colour only represents levels of recreation intensity locally (Robb, 2017). At our Heatmap zoom level of 12 (19 m resolution), heat map values are comparable within an approximate 50 km diameter, similar to previous studies (Corradini et al., 2021). Our results suggest that the Strava Heatmap may be more appropriate to use for smaller extents, such as within our case study areas.

Our findings highlight how application-based tools provide an incomplete representation of recreation because they capture the relative popularity of recreation in areas among app users. For instance, in the Ghost Public Land Use Zone (PLUZ) case study, we found that application-based Wikiloc data did not capture all summer motorized trails. Moreover, application-based data exhibits unequal contribution of data among users because the most active users contribute more than others and is often biased towards demographic groups (Ghermandi & Sinclair, 2019; Venter et al., 2023). These comparisons highlight how the spatial and temporal distribution of application-based data are dependent on the number and types of users on the platform. This may be further compounded by changing app usership over time (Venter et al., 2023). While application-based tools can provide detailed data about recreation for large areas, the data may not reliably address spatial or temporal recreation patterns if biases in app use are not properly accounted for. As we demonstrate here, cross-referencing application-based data with traditional tools offers a solution for identifying these biases and improving the accuracy of recreation estimates.

In contrast to the application-based data from Strava products and Wikiloc, PM and aerial surveys provided information on where recreation was occurring regardless of activity or app use (Fig. 6). Our PM data was collected at a coarse spatial scale, making it challenging to directly compare with spatiotemporal data from cameras, counters, and GPS-track data from Strava Metro or Wikiloc across the study area. Similarly, the aerial survey data was difficult to compare with data from other tools because it represented a single snapshot of recreation in time. Although multiple surveys can be conducted throughout the season to increase the temporal coverage of aerial survey data, the cost and time for aerial surveys limits how frequently this type of data can be collected. An advantage of Wikiloc, PM and aerial surveys is that these tools collect off-trail recreation data without having prior knowledge of where recreation occurs. Moreover, aerial surveys capture areas with low intensity of use that other tools did not (Fig. 8) and when combined with other tools can provide more robust and detailed recreation information.

Study limitations

While our work provides insight into the application of traditional and application-based tools for monitoring recreation, our results need to be interpreted in light of the study’s limitations and scope. Our datasets were collected from various wildlife and recreation monitoring projects, resulting in a spatial and temporal mismatch between datasets. Future studies would benefit from having tools with the same spatial and temporal coverage to allow for direct comparisons of tools across greater areas and time periods. Despite this, our approach of leveraging multiple datasets from existing projects reflects data collection limitations often faced by managers due to time, funding, and staffing constraints. Further, this approach demonstrates how supplementing data from other projects can alleviate the costs and labour associated with data collection and, in many cases, can enhance the temporal and spatial extents and sample size of data.

Future research and opportunities

Our study highlights additional gaps in knowledge. Our work focused on trail based recreation. However, recreationists continue to venture off trail and further into the backcountry. Our research on aerial surveys and Wikiloc adds to previous research capturing off-trail recreation (Norman & Pickering, 2017; Heinemeyer et al., 2019). Due to the limited availability of datasets, we only assessed Wikiloc’s ablity to capture motorized recreation in specific locations. Because Wikiloc may reflect recreationists in more remote areas (Norman & Pickering, 2017) and captures a variety of recreation activities, future work should assess Wikiloc’s ability to quantify off-trail use. However, use of this tool should incorporate careful consideration of biases and constraints of manually downloading individual Wikiloc tracks (Goodbody et al., 2021). PM holds promise for identifying broad areas with informal recreation trails and activities not readily captured by other tools, such as winter and water-based recreation. Information from PM could be used to strategically select areas for ground or aerial surveys, cameras or counters, or prioritize areas for collecting application-based data. However, many user-generated platforms are regularly updated (i.e., format and data collected) that can cause challenges when using this data for multi-year studies and requires ongoing assessment for limitations and biases. For instance, our Strava Heatmap data from 2016 to 2017 categorized recreation into broad categories (run, hike, walk, ride, water, winter), but updates categorized recreation into more specific activities (i.e., backcountry skiing, alpine skiing, snowshoeing, kayaking; see Table S2). While this updated dataset has potential for answering recreation-specific research questions, further research should assess limitations and biases of this more detailed data.

Numerous networks have been formed to facilitate the use of monitoring tools and data sharing and collaboration in ecology research (e.g., Data Basin; WildCAM; WildTrax; Steenweg et al., 2017). Development of specific human recreation communities of practice (e.g., https://wildcams.ca/) would help standardize and centralize recreation data and promote data sharing to improve monitoring and management. Further, collaboration and standardization of monitoring and data practices would help address the challenges of insufficient staff and financial resources limiting the adoption of application-based data for recreation monitoring and management (Mangold et al., 2024).

Implications and recommendations for management

Identifying tools that provide accurate and detailed data on recreation use is the foundation of recreation monitoring and planning (Cessford & Muhar, 2003; D’Antonio et al., 2012). Selection of the use of a single, or combination of tools, should be based on an understanding of the strengths and limitations of each tool relative to clearly defined management and planning objectives and priorities (Cessford & Muhar, 2003). We provide a summary table (Table S2) highlighting key strengths and limitations of the tools we assessed.

Evidence-based decisions often require detailed information about recreation use and spatial patterns. For managers seeking to quantify fine spatial and temporal patterns of on-trail recreation within a park or trail network, cameras and counters can provide accurate and detailed information. The decisions to use camera traps or counters should be based on context-specific considerations. For instance, if managers require information on the type of recreation activity and group sizes, cameras should be used rather than counters because the latter lacks the capacity to differentiate across all recreation types and group sizes (Table S2). While cameras provide detailed data on recreation types and number of recreationists, this tool requires consideration of human privacy and is resource intensive (e.g., replacing batteries, processing images, but see Fennell, Beirne & Burton (2022) for details on automatic classification software to improve image processing efficiency). It is also worth noting that cameras can monitor wildlife and population trends, potentially facilitating research on human recreation-wildlife interactions (Steenweg et al., 2016; Naidoo & Burton, 2020; Barrueto et al., 2022). Comparatively, data processing for counters is generally quick and is less labour intensive (Miller, Leung & Kays, 2017). However, counters systematically underestimate recreation intensity when groups of people trigger the counter only once (Miller, Leung & Kays, 2017; Lawson, 2021) and can overestimate recreation intensity when triggered by animals or vegetation. However, placement of counters on narrow trail sections where people are more likely to travel single-file and calibration to adjust counts to reflect estimates of the actual number of recreationists can help address this challenge (Cessford & Muhar, 2003). We suggest that pairing tools, such as counters with cameras, can derive additional data, allowing for cross-validation and more accurate estimates.

Application-based tools can supplement traditional tools and provide detailed information across large landscapes, although these platforms vary on the types of human activities reflected in their data. Use of application-based tools must be carefully considered, with biases in spatial extents and user groups addressed (Table S2). We suggest that application-based data be supplemented with traditional tools or data from regional or large-scale monitoring efforts already underway to improve confidence in estimates.

Alternatively, when recreation data at fine spatial resolutions is not needed, but extent is important, aerial surveys, PM and Strava Heatmap offer alternative approaches to collect coarser recreation data. Strava Heatmap includes off-trail winter and water activities in addition to hiking, walking, running and biking. However, the information from this platform is at a coarse time scale making it challenging to integrate with higher resolution data from cameras or counters. Further, Strava Heatmap may not provide accurate estimates for areas with very low use. Similarly, aerial surveys can capture off-trail winter recreation, both extent and index of use. These surveys can be used to monitor changes over time in a repeatable methodology. However, weather, cost and labour may limit aerial surveys spatial and temporal extents. Alternatively, PM can be designed to provide information on intensity of use and recreation activity type with the desired spatial or temporal resolution to meet manager’s needs. However, the quality of PM data is sensitive to sampling design, participants cognitive bias and their characteristics, knowledge of place, and experiences that shape perspectives (Brown, 2011; Brown & Kyttä, 2014). Collectively, PM and Strava Heatmap can provide baseline information on where recreation may occur and can be used for prioritizing areas for placement of traditional tools for more intensive monitoring. The use of Strava Heatmap, aerial surveys and PM may be particularly helpful for cases when employing large networks of cameras or counters is impractical, such as monitoring off-trail recreation in large and rugged landscapes or across multiple recreation areas.

While incorporation of application-based tools for monitoring is likely to increase in the future (Mangold et al., 2024), digitalization presents challenges and opportunities. Many recreationists use digital outdoor and fitness platforms for planning and navigating activities. These tools can be used for visitor management (i.e., promoting official trails, adding trails information on crowdsourced map-services such as OpenStreetMaps; Mangold et al., 2024). However, unofficial trails (i.e., trails not managed for recreation) are frequently included in these platforms and are subsequently shared with many users (Mangold et al., 2024; Schwietering et al., 2023). Many of these unofficial trails are not included in government databases (Loosen et al., 2023). The lack of information about which recreation activities are occurring on unofficial trails is a challenge; managing human-wildlife conflict and conflict between recreation users is difficult if the extent of recreation and types of recreation occurring is unknown. Leveraging information from application-based tools can provide managers with additional information to validate their current databases.

Conclusions

The increase in outdoor recreation has come with a demand for tools that accurately estimate the extent and intensity of recreation. This information constitutes the basis for understanding the impacts of recreation on wildlife and natural areas, developing evidence-based land use management plans and promoting enjoyable and sustainable recreation. Yet, the access to timely, accurate, and detailed recreation data is not keeping pace to the expanding recreation footprint. We addressed this knowledge gap by comparing the use of traditional and application-based tools for measuring recreation across a large extent. We show that recreation use can be estimated using traditional and application-based tools, although their accuracy and utility varies across space, time and activity type. We recommend that the specific context and management objectives guide selection of tools. Application-based data from apps should be supplemented with data from traditional tools to identify individual tool biases and improve confidence in estimates. Our research can help managers select which tools to use to improve recreation monitoring. This information can guide decisions to better protect ecological systems while allowing for sustainable recreation.

Supplemental Information

Supplemental Information 1 R code to run analyses and figures

Supplemental Information 2 Aerial survey and participatory mapping data

Supplemental Information 3 Number of pairwise combinations of camera, counter, and Strava Metro spatially matched locations for all recreation activities, biking, and pedestrian (hike, run, walk)

Cameras and counters were matched if devices were within 200 m of each other along a trail and had data on the same day. Cameras and counters were matched to Strava Metro segments within 30 m of devices and had data on the same day.

Supplemental Information 4 Comparison of the select monitoring tools to measure motorized (M) and non-motorized (NM) winter and summer outdoor recreation, including key limitations and advantages for each tool (does not represent an exhaustive list)

Supplemental Information 5 Data Description and Supplemental Figures

Supplemental Information 6 Monthly counts of pedestrian and biking recreationists from spatially matched locations of cameras, counters and Strava Metro

Each panel represents monthly pedestrian counts between spatially matched locations of (A) cameras and counters, (B) cameras and Strava Metro, (C) counters and Strava Metro, and monthly biking counts from spatially matched locations of (D) cameras and counters, (E) cameras and Strava Metro and (F) counter and Strava Metro from 2017 –2020. The red line represents a trendline generated with generalized additive models.

Supplemental Information 7 Pearson’s correlation values of monthly counts of all (row A) recreation activities, (row B) pedestrians, (row C) biking for pairwise combinations of spatially matched cameras, counters and Strava Metro locations

First column compares counts from cameras and counters; second column compares counts from cameras and Strava Metro; third column compares counts from counters and Strava Metro. Base map source: Esri, HERE, Garmin, Intermap, increment P Corp., GEBCO, USGS, FAO, NPS, NRCAN, GeoBase, IGN, Kadaster NL, Ordnance Survey, Esri Japan, METI, Esri China (Hong Kong), ©OpenStreetMap contributors, and the GIS User Community.

Supplemental Information 8 This research explores the use of traditional and application-based tools for measuring human outdoor recreation in western Alberta and eastern British Columbia, Canada, and within the territories of the Okanagan/Syilx, Sinixt, Ktunaxa, Secwépemc, Niitsit

It also includes lands within Treaties 6, 7, and 8 and districts 1, 3 and 4 of the Otipemiskiwak Métis Government of Alberta. Credits: Loosen Studio (loosenstudio.net).

We thank all data partners (Ministry of Forestry, Parks and Tourism, Government of Alberta; Nature Conservancy of Canada; Parks Canada Agency; Recreation Sites and Trails British Columbia) for sharing data for this research and for the insight and feedback that guided earlier drafts of this paper. Special thanks to Naia Noyes-West for leading the participatory mapping and Kim Heinemeyer and Peggy Holroyd for thoughtful feedback and engaging discussion that helped shape this research. We also thank Loosen Studio for creating visual representations of our research. Thank you to the journal editor and two reviewers for their constructive comments.

Additional Information and Declarations

Competing Interests

Author Contributions

Data Availability

The authors declare there are no competing interests.

Talia Vilalta Capdevila conceived and designed the experiments, performed the experiments, analyzed the data, prepared figures and/or tables, authored or reviewed drafts of the article, and approved the final draft.

Brynn A. McLellan conceived and designed the experiments, performed the experiments, analyzed the data, prepared figures and/or tables, authored or reviewed drafts of the article, and approved the final draft.

Annie Loosen conceived and designed the experiments, authored or reviewed drafts of the article, and approved the final draft.

Anne Forshner conceived and designed the experiments, authored or reviewed drafts of the article, contributed data, and approved the final draft.

Karine Pigeon conceived and designed the experiments, authored or reviewed drafts of the article, project supervision, and approved the final draft.

Aerin L. Jacob conceived and designed the experiments, authored or reviewed drafts of the article, project supervision, and approved the final draft.

Pamela Wright conceived and designed the experiments, authored or reviewed drafts of the article, project supervision, and approved the final draft.

Libby Ehlers conceived and designed the experiments, authored or reviewed drafts of the article, project supervision, and approved the final draft.

The following information was supplied regarding data availability:

Data from Strava Global Heatmap (https://www.strava.com/maps) and Wikiloc (https://www.wikiloc.com) are available upon creating a user account.

Detailed descriptions of these trail camera, infrared counters and Strava Metro data sets and aggregation of these data sets are available in the article text and Data Description S1. The aerial survey data and participatory mapping data supporting the findings of this study are available in the Supplemental File.

The trail camera data, infrared counter data, and Strava Metro data are not openly available due to sensitivity and license agreements from data contributors. Ownership of this data resides with the following researchers and agencies and data can be made available upon request and possible license agreements. Trail camera data are available from: Banff National Park, Parks Canada Agency (banffinfo@pc.gc.ca); Forests, Parks and Tourism, Government of Alberta (AEP.ParksEcology@gov.ab.ca); Mount Revelstoke and Glacier National Parks, Parks Canada Agency (mrg.information@pg.gc.ca). Infrared counter data are available from: Canadian Rocky Mountains, Nature Conservancy of Canada (canadian.rockies@natureconservancy.ca); Forests, Parks and Tourism, Government of Alberta (AEP.ParksEcology@gov.ab.ca); Kootenay National Park, Parks Canada Agency (kootenayinfo-infokootenay@pc.gc.ca); Mount Revelstoke and Glacier National Parks, Parks Canada Agency (mrg.information@pg.gc.ca); Recreation Sites and Trails British Columbia, Government of British Columbia (https://www.sitesandtrailsbc.ca/contact-us.aspx). Strava Metro data are available by applying for a Strava Metro partnership (https://metroview.strava.com/application; email address for support support@stravametro.zendesk.com). This partnership provides access to the Strava Metro platform where data can be downloaded. The R code to run analyses is included in the Supplemental File.

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
