# Peer review of "Advancements in monitoring: a comparison of traditional and application-based tools for measuring outdoor recreation"

_PeerJ, doi:10.7717/peerj.17744_

## Round 0.1 · original submission · Minor Revisions

Both reviewers I have consulted found the paper interesting and well written. They had several comments and requested minor revisions before the paper can be further considered for publication. Please ensure that all the points they raised are adequately addressed in a revised version of the manuscript.

·

Basic reporting

• Introduction and discussion are both very well written and clear – sections that can be hard to write well for a paper and topic this large.
• Paper is structured well and includes relevant citations.
• Check your section numbering throughout – e.g., there is a section 4.2 but no section 4.1, there are two sections labeled 4.3, there is a 3.1.1 but no 3.1 (maybe third level headings there should be second-level)?
• I can’t seem to find figures S4 and S5 referenced in the text. Or S1? Double check that all your supplementary files are referenced in the text.
• None of the supplementary figures have captions. Is this normal for this journal? I had a really hard time understanding some of them without captions (e.g., I have no idea what S4 is showing).
• For some additional specific suggestions to increase clarity, see my comments under “additional comments”

Experimental design

• This is original primary research within the journal scope.
• The methods are rigorous and defensible. The methods are described with enough detail for the reader to understand.
• I really like how you list all your research questions in Table 1, it’s very clear and they are well defined. One suggestion – you may want to number these or refer back explicitly to these in the results, because this paper has so much going on that it can get hard to follow as a reader.

Validity of the findings

• Data were provided to reproduce the results – data appear to be robust.
• I ran the R code and everything worked as expected.
• Conclusions do match what was found in the results.

Additional comments

I reviewed the paper titled “Advancements in monitoring: a comparison of traditional and application-based tools for measuring outdoor recreation.” This is a large-scale paper that compares various approaches to estimating recreational visitation using different methods/datasets at many locations in the Canadian Rockies. I commend the authors for pulling together data from so many different sites. This paper is a valuable contribution to the academic literature, and will also help researchers better communicate the limitations of some of these app-based tools to land managers. For instance, I know in the U.S. there is a lot of interest in using the Strava datasets, and I appreciate that this research clearly shows how it compares to other data and the possible biases and limitations.
Additional specific comments are below:

• When you mention “counters” In the abstract, it may be helpful to specify if you are referring to infrared counters, as there are many different types of counters.
• Line 225 – I’d recommend adding in the PM section the total number of people you talked to. Also, how did you come up with the final numbers? For instance, did you take the mean of what each participant indicated?
• Line 298 – isn’t the temporal scale of Strava heatmaps 13 months? Can you rephrase this for clarity as to why a 9-month period would match the Strava heatmaps scale?
• On the caption of figure 1 – your reference to “PLUZ, panel A” I think is supposed to reference panel B instead.
• On the caption of figure 4 – you mention panels A, B, and C, but not the other 6 panels. I believe the reference to “all activities” for instance should be panels A, D, and G and not only A, is that correct?
• On figure 6, panel A (and figure 8, panel C)– does this represent the estimated average people per day, month, or other timescale? Might be useful to indicate it in the figure itself so the figure can be interpreted without going back to the methods section.
• Throughout the paper (and in supp table 2), the authors mention that counters can’t distinguish between individuals or groups. Yes, but isn’t that why it is advised to put the counters in a narrow section of trail where people are more likely to be walking single file (if possible)? And missing some people that walk double-file should be able to be adjusted for if using calibration techniques properly. It feels like some of this nuanced could be mentioned somewhere.
• Supplementary table S2 – you indicate that Strava heatmap can detect off trail use. Is that really correct? My understanding is that Strava heatmap snaps to OpenStreetMap roads or trails, so I’m not entirely sure how that could detect off trail use. You indicate that Strava Metro is snapped to OSM, but isn’t the heatmap also snapped to OSM? I could of course be wrong - worth double checking though and maybe mentioning somewhere in the methods if Strava Metro is snapped to OSM but Strava Heatmap is not.
• Supplementary table S2 – why are there no advantages listed for counters? There are definitely some advantages, assume this was just a mistake.

Reviewer 2 ·

Basic reporting

Overall, the paper was well written and represents a valuable contribution to our understanding of how various tools can be used alone or in combination to effectively estimate recreation activity.

There is a lot in this paper - specifically, there are many comparisons with various datasets and tools. This is a strength of the paper but, at the same time, it can be hard to follow at times. While I see the value in these tests I suggest that the authors make sure the inclusion of each is clearly justified. For example, the reason for including the Strava Heat Maps isn't clear, since Strava Metro is also included, besides that it is stated that the latter is already used to measure recreation activity. I'd also like to see, either in the Intro or Methods, why these species tools were chosen for inclusion in this study, other than each belonging to one of the two broader categories.

One suggestion the authors may wish to consider is, instead of having headings for the case studies, that they lead this instead with what comparisons are being made. E.g. 1st heading in Results something like "comparison of all methods" then instead of "Case Studies", could be "Comparison of X methods" etc. Since this is more of a 'methods' paper, the place of where these data come from are of secondary importance. I.e. we don't really about what the intensity of biking is, we care about how closely the methods give similar (or different) type of information.

Experimental design

No comment

Validity of the findings

Given that there were several comparisons made in this paper with the use of various tools, it may be helpful to include a table with some general recommendations, to compile/consolidate some of the nuances associated with the recommendations and management implications. This would help reduce the amount of text in the discussion and make and help make the paper easier to follow.

Additional comments

Minor comments

Abstract
L33: What do you mean by ‘traditional’ and ‘application-based? If possible, define here.
L39-45: I suggest moving this part up where you describe what you mean by traditional vs application based

Intro
89-91- example? It’s a bit confusing that traditional is mentioned 1st here but then application is discussed in greater detail. Then the next paragraph does into details about traditional. Suggest restructuring this to improve the flow.

Methods
L209- does this represent Jan 1 2017 to Dec 31 2019?
L214, 225- for each of these methods can you include a reference that also used similar approach?

L277- justification of these distances chosen? Any reference for 200m?

Results
L333- what is ‘high value’ just highest values of categories? i.e. for raster

---

## Round 0.2 · accepted · Accept

I am glad that the data sharing issues could be solved to everyone's satisfaction and that this interesting paper can move forward. The minor revisions requested by the reviewers have been addressed adequately and I am glad to accept the manuscript for publication. Congratulations to the authors!